# Microplastics Exacerbate Cadmium-Induced Kidney Injury by Enhancing Oxidative Stress, Autophagy, Apoptosis, and Fibrosis

**DOI:** 10.3390/ijms232214411

**Published:** 2022-11-19

**Authors:** Hui Zou, Yan Chen, Huayi Qu, Jian Sun, Tao Wang, Yonggang Ma, Yan Yuan, Jianchun Bian, Zongping Liu

**Affiliations:** Jiangsu Co-Innovation Center for Prevention and Control of Important Animal Infectious Diseases and Zoonoses, College of Veterinary Medicine, Yangzhou University, Yangzhou 225009, China

**Keywords:** microplastic, cadmium, kidney, oxidative stress, autophagy

## Abstract

Cadmium (Cd) is a potential pathogenic factor in the urinary system that is associated with various kidney diseases. Microplastics (MPs), comprising of plastic particles less than 5 mm in diameter, are a major carrier of contaminants. We applied 10 mg/L particle 5 μm MPs and 50 mg/L CdCl2 in water for three months in vivo assay to assess the damaging effects of MPs and Cd exposure on the kidney. In vivo tests showed that MPs exacerbated Cd-induced kidney injury. In addition, the involvement of oxidative stress, autophagy, apoptosis, and fibrosis in the damaging effects of MPs and Cd on mouse kidneys were investigated. The results showed that MPs aggravated Cd-induced kidney injury by enhancing oxidative stress, autophagy, apoptosis, and fibrosis. These findings provide new insights into the toxic effects of MPs on the mouse kidney.

## 1. Introduction

Cadmium (Cd) is a highly toxic heavy metal in the environment. Its harm to human health mainly comes from environmental pollution caused by industrial and agricultural production. Cadmium has a long biological half-life and low excretion rate in the human body [1]. The kidney is one of the most important accumulation sites and target organs of Cd. Chronic Cd exposure causes renal damage, and this process is generally considered irreversible [2,3]. There is no effective treatment available, and the number of deaths from kidney damage caused by Cd exposure is reported to be increasing every year [4,5]. Therefore, it is necessary to clarify the relationship between Cd in the environment and kidney damage.

Microplastics (MPs) are plastic particles with a diameter of less than 5 mm [6] and are major pollutants. Due to plastic’s debris durability, these particles are ubiquitous in the terrestrial and marine environment [7]. NPs and MPs were detected in global surface water, continental waters, soil, atmosphere, and even polar regions [8]. Microplastics might cause unpredictable adverse effects on human health through food chain transport [8,9]. Microplastics have been reported to accumulate in the intestine, liver, and kidneys of mice [10,11]. The kidney can excrete ultrafine particles, while the deposition and accumulation of MPs in the kidney will cause damage, including physical damage and an immune response [12,13,14]. A study showed that microplastics (4 μm in diameter) could be found in the intercellular space and renal tubule using transmission electron microscopy, and the number of 4 μm particles entering the tissue was lower than that of 600 nm particles [15]. In that study, the organ indices and serum biochemical parameters of the kidney varied more significantly than those of other organs, such as the liver and heart [15]. Therefore, the kidney is an important target organ to study the effects of MPs on mammals (unlike aquatic animals). However, the toxic effects of MPs on mammalian kidneys remain unclear and require further study.

Furthermore, MPs in the environment can serve as vectors for organic and metallic contaminants owing to their remarkable binding capacity [16,17]. According to a report on the soil contamination status in China issued in 2014, Cd ranks first among the metallic contaminants in the farmland exceeding Chinese soil environmental quality standards [18]. Recent studies have confirmed the occurrence of various MPs in farmlands [19], as well as in freshwater [20]. Microplastics and the heavy metals adhering to their surfaces can be taken up by organisms, leading to heavy metal deposition. Microplastic ingestion provides a pathway for the transfer of biological contaminants such as heavy metals into organisms through ingestion [21]. Experiments have shown that exposure of fish to MPs and Cd, either singly or in combination, for 30 days resulted in an increase in MP concentrations that reduced Cd accumulation in the fish. Co-exposure led to severe oxidative stress, which could stimulate innate immunity [22]. Although there have been many studies on the effects of MPs and heavy metals on aquatic animals, there have been few studies on the combined effects of MPs and Cd on mammals.

Studies have shown that co-treatment with MPs with Cd enhances early growth, oxidative stress, and apoptosis in zebrafish, resulting in tissue damage [23]. In addition, MP-induced myocardial dysplasia in birds is mainly attributed to the endoplasmic reticulum stress-mediated autophagic pathway [24]. These results suggest that the effects of exposure to plastic particles and Cd contamination are complex, with toxic effects varying among organisms. Furthermore, studies on the exposure of MPs and Cd in mammals and their potential toxicity mechanisms are still limited. Therefore, further study is needed to determine the toxic effects of co-exposure of MPs and Cd on mammals.

## 2. Results

### 2.1. Microplastics Aggravate Cadmium-Induced Kidney Damage

Mice were treated with Cd (50 mg/L) and/or 5 μm MPs (10 mg/L) for 90 days. During the whole study, no abnormal behavior or symptoms were observed in any group of mice. The food intake and body weight of the mice were monitored daily, and the organ coefficients decreased more significantly in the mice in the Cd and MP co-treatment group than in either single treatment group (Figure 1A,B). To investigate the role of Cd in kidney damage with MPs, we assessed the histological changes in the kidneys using H&E staining (Figure 1C). Compared with that in the control group, Cd resulted in kidney tissue with more visible dilatation of the renal capsule lumen (black arrow), necrosis and detachment of the tubular epithelium (blue arrow), cytoplasmic sparing and light staining, and increased erythrocyte (red arrow) and monocyte infiltration (green arrow) around a few vessels. These findings confirmed that the model of Cd-induced nephrotoxicity had been successfully established. In addition, co-treatment with MPs and Cd significantly increased the Cd-induced tubular injury, with occasional dilatation of the renal peritoneal lumen (yellow arrow). In addition, we examined the renal ultrastructure using transmission electron microscopy. As shown in Figure 1D,E, in the control group, the nucleus was intact and the mitochondrial structures and mitochondrial cristae were clearly visible. However, the Cd group showed significant ultrastructural changes in terms of nuclear depression and deformation (red arrow). In addition, Cd also induced mitochondrial damage, as evidenced by significant vacuolization (blue arrow) and cristae disruption (black arrow). The nuclear and mitochondrial damage were increased in the co-treatment group. In addition, more lipid droplets (green arrow) were observed in the co-treatment group, and it is suspected that lipid accumulation occurred, which could be further investigated. In addition, we assessed the accumulation and localization of fluorescent polystyrene microspheres in kidney tissue. The results showed that the microspheres accumulated in the mouse renal tubular epithelium, suggesting that microplastics with a 5 μm particle size can enter the kidney (Figure 1F). Taken together, these results confirmed that MPs exacerbated Cd-induced kidney injury in vivo.

### 2.2. Microplastics Exacerbate Cd-Induced Oxidative Stress in Mouse Serum and Kidneys

Mice were treated with Cd (50 mg/L) and/or 5 μm MPs (10 mg/L) for 90 days. To explore whether MPs can enhance Cd-induced systemic and renal oxidative stress in vivo, we assessed the oxidative stress parameters in the serum and kidney in the mouse model. The results showed that Cd exposure significantly elevated the serum MDA levels and decreased CAT, GSH, and SOD levels compared with those in the control group (Figure 2A–D). However, the co-treatment group had significantly higher MDA levels and no significant changes in CAT, GSH, and SOD levels compared with those in the Cd group. In addition, we examined the levels of oxidative stress-related proteins. Western blotting showed that Cd exposure significantly increased the levels of SOD2, HO-1, and Sirt3 in the kidney compared with those in the control group. Compared with those in the Cd group, SOD2 and Sirt3 levels were significantly elevated in the co-treatment group, while the HO-1 levels did not change significantly (Figure 2E–H). These results confirmed that co-treatment with MPs and Cd increased the levels of systemic and renal oxidative stress in mice in vivo.

### 2.3. Effect of Co-Treatment with MPs and Cd on Autophagy in Mouse Kidneys

Mice were treated with Cd (50 mg/L) and/or 5 μm MPs (10 mg/L) for 90 days. To investigate the effect of MPs and Cd on the level of autophagy in the kidney, we detected autophagy-related proteins by western blotting and observed the number of autophagosomes using transmission electron microscope. Western blotting showed that, compared with those in the Cd group, the levels of the autophagy marker LC3 and the early autophagy proteins ATG5, Beclin-1, and ATG7 were significantly higher in the co-treatment group (Figure 3A–E). In addition, the number of autophagosomes in the co-treatment group was higher than that in the Cd group (Figure 3F). These results suggest that the co-treatment with MPs with Cd increased the level of autophagy in mouse kidneys in vivo.

### 2.4. Effect of Co-Treatment with MPs and Cd on Apoptosis in Mouse Kidneys

Mice were treated with Cd (50 mg/L) and/or 5 μm MPs (10 mg/L) for 90 days. To investigate the effect of MPs and Cd on the level of apoptosis in the kidney, we detected apoptosis-related proteins by IHC and western blotting. The IHC results showed that compared with those in the Cd group, the areas that were positive for Bax, Cytc, and Caspase-3 were significantly increased in the co-treatment group (Figure 4A–D). Western blotting showed that the Bax/Bcl-2 ratio was increased in the co-treatment group compared with that in the Cd group, and the expression levels of their downstream regulatory proteins, Caspase-3 and Cleaved Caspase-3, were both significantly increased, whereas the levels of Cleaved Caspase-9 did not change significantly (Figure 4E–I), which was consistent with the IHC results. These results confirmed that co-treatment with MPs and Cd increased the level of apoptosis in mouse kidneys in vivo.

### 2.5. Effect of Co-Treatment with MPs and Cd on Kidney Fibrosis in Mice

Mice were treated with Cd (50 mg/L) and/or 5 μm MPs (10 mg/L) for 90 days. To investigate the effect of MPs and Cd on the extent and severity of renal fibrosis, we assessed the level of renal fibrosis using Sirius Red and Masson staining. Both staining results showed a significant increase in collagen fibers in the kidney tissue of mice in the MPs and Cd co-treatment group compared with that in the Cd group (Figure 5A–C). In addition, we examined the levels of the fibrosis marker proteins α-SMA, TGF-β1, and COL4A1 (Figure 5D–G). Western blotting showed that the levels of α-SMA and COL4A1 were significantly higher in the co-treatment group than in the Cd group, while no significant differences were observed in the levels of TGF-β1. The above results indicated that MPs exacerbated Cd-induced renal fibrosis in vivo.

## 3. Discussion

Cd is a non-degradable heavy metal in the environment that is relevant to human health and industrial production. There is growing evidence that renal damage is a major feature of Cd-induced toxicity [25]. Cd causes nephrotoxicity through different mechanisms, such as oxidative stress [26], autophagy [27], apoptosis [28], and fibrosis [29]. Microplastics can attract other polluting particles from the surrounding environment and the combined effect can increase their contamination, posing a potential hazard to the ecological environment [30]. The toxicity of microplastic sorption depends on the nature of the sorbent, the particle size and the composition of the microplastic [31]. Considering the intrinsic link between MPs and heavy metal toxicity, this study aimed to evaluate nephrotoxic effects of MPs on Cd-induced nephrotoxicity.

The changes in body weight and organ coefficients of experimental animals can reflect the toxic effects of toxins on animal bodies and organs. H&E staining and transmission electron microscopy can reveal kidney damage at the ultrastructural level. fluorescent polystyrene microspheres allow observation of the accumulation and localization of MPs in kidney tissue. The results for the co-treatment group showed a highly significant decrease in kidney coefficients compared with those in the control group. H&E staining in the Cd group showed dilatation of the renal capsule lumen, necrosis and detachment of the tubular epithelium, and infiltration of mononuclear cells. Transmission electron microscopy showed nuclei degeneration, mitochondrial vacuolation, and cristae disruption. It was determined that 5 μm MPs accumulated in the mouse renal tubular epithelium and entered the renal tissue, which is consistent with previous findings [32]. These results suggested that the Cd-mediated kidney injury model was successfully established and that MPs co-treatment enhances its deleterious effects.

One of the most important causes of kidney injury is oxidative stress caused by excessive production of reactive oxygen species (ROS) and inhibition of the antioxidant system [33,34]. In a laboratory experiment, early juveniles of Symphysodon aequifasciatus (discus fish) were exposed to single and combined effect of polystyrene microplastic (0, 50, or 500 mg/L) and cadmium (0 or 50 mg/L). Co-exposure resulted in severe oxidative stress and could stimulate innate immunity; however, there was no effect on growth and survival rate of the juveniles [31]. Earlier studies showed that Cd promotes ROS production in renal cells through a variety of indirect mechanisms, such as disruption of electron flow in the mitochondrial respiratory chain, transition metal release, and inactivation of endogenous redox defense molecules through binding to sulfhydryl groups [35,36]. Elevated MDA levels have also been reported as an indicator of lipid peroxidation [37]. As protection against oxidant stress, the antioxidant enzymes SOD, CAT, and GSH are widely considered as powerful defense mechanisms against oxidative kidney damage [38]. In this study, Cd or MPs caused significant oxidative stress, as evidenced by increased MDA levels and decreased antioxidant enzyme activities (SOD and CAT) and GSH levels in serum. However, apart from increasing MDA levels, co-treatment with MPs and Cd did not further decrease SOD, CAT, and GSH activities, which might be a response to the excess ROS stimulated by them in cells [6]. Research has shown that SOD2 is the downstream mediator of Sirt3, protecting the nucleus and mitochondrial DNA, as well as other cellular macromolecules, from ROS-related damage through its attenuation of ROS levels [39,40]. The results of the present study showed that MPs significantly upregulated Sirt3 and SOD2 levels following Cd exposure. The different effects on Sirt3 levels might be related to different stages of the body’s tolerance to oxidative stress. In conclusion, these data suggest that MPs play an important role in Cd-induced oxidative stress in mouse kidneys.

Autophagy is a dynamic process in which damaged macromolecules and organelles within cells are degraded and recirculated to synthesize new cellular components. Recent studies have begun to reveal the role of autophagy in progressive kidney disease and subsequent fibrosis. In this study, MPs increased the levels of Cd-mediated autophagy marker LC3-II and the autophagy early proteins ATG5, Beclin-1, and ATG7 and increased the levels of the renal fibrosis marker proteins α-SMA, TGF-β1, and COL4A1. This suggests that MPs play an important role in Cd-mediated renal autophagy and fibrosis.

Apoptosis is a controlled form of cell death characterized by cell shrinkage, chromosome condensation, and breakage [41]. The cell death receptor (exogenous) and mitochondrial (endogenous) pathways are the two main signaling pathways that trigger apoptotic cell death. It is reported that in mitochondrial pathway-mediated apoptosis, Caspase-9 can be activated to induce apoptosis by initiating the downstream Caspase-3 pathway. In addition, Bax and Bcl-2 were the first proteins identified to be involved in the positive and negative regulation of apoptosis. Bax induces apoptosis when it forms a homodimer; whereas, when Bax forms a heterodimer with Bcl-2, it activates the function of Bcl-2 to inhibit apoptosis. Both of these pathways contribute to oligomerization of pro-apoptotic proteins in mitochondria and induce the release of mitochondrial cytochrome c [42]. In this study, we found that MPs significantly upregulated Caspase-3, Cleaved Caspase-3, and Bax levels and downregulated Bcl-2 levels following Cd exposure. In addition, the IHC results confirmed that mitochondrial cytochrome c was released into the cytoplasm. In conclusion, these data suggest that MPs play an important role in Cd-induced apoptosis in mouse kidney cells.

Previous studies showed that oxidative stress affects the production and balance of autophagy, apoptosis, and fibrosis, thereby exacerbating tissue damage [43,44,45]. Therefore, further studies on autophagy, apoptosis, and fibrosis might confirm the combined toxicity of MPs and Cd. Autophagy degrades long-lived cytoplasmic proteins and organelles, provides substrates for energy metabolism, and restores amino acids, fatty acids, and nucleotides to meet the needs of cellular biosynthesis. In the present study, combined exposure to MPs and Cd resulted in elevated levels of autophagy. This might have occurred through excessive accumulation of ROS, which induces sustained autophagy in the mouse kidney [46]. However, further experimental studies are needed to confirm this. Recent studies have also begun to reveal the role of autophagy in progressive nephropathy and subsequent fibrosis. Livingston et al. (2016) first reported the pro-fibrotic role of autophagy in the mouse kidney. Studies showed that overexpression of WNT1-inducible signaling pathway protein 1 (WISP-1) increased LC3-II and Beclin-1 expression and exacerbated renal fibrosis in the unilateral ureteral obstruction (UUO) model and in TGF-β-treated renal tubular epithelial cells [47]. In this study, we hypothesized that there might be an association between sustained autophagy and exacerbation of fibrosis in the mouse kidney; however, further studies are needed. Apoptosis is a tightly regulated intracellular program in which cells are destined to die and is activated to degrade cellular DNA, nuclear proteins, and cytoplasmic proteins [48]. In this study, combined exposure to MPs and Cd induced excessive apoptosis, which might lead to normal tissue cell death, possibly inhibiting tissue repair and thus exacerbating tissue damage [49,50]. In this model, excessive autophagy might lead to apoptosis. The above results suggest that oxidative stress induced by combined exposure to MPs and Cd affects the levels of autophagy, apoptosis, and fibrosis, which leads to further kidney damage in mice. However, further experimental studies are needed to confirm this.

## 4. Materials and Methods

### 4.1. Chemicals and Reagents

Cadmium chloride (CdCl_2_) was purchased from Sigma Aldrich (St. Louis, MO, USA). Polystyrene fluorescent microspheres (5 μm; 7-1-0500) and polystyrene microspheres (5 μm; 6-1-0050) were purchased from BaseLine Chromatographic Technology Development Center (Tianjin, China). All antioxidant enzyme detection kits were obtained from Nanjing Jiancheng Bioengineering Institute (Nanjing, China). The following primary antibodies were used: Anti-BCL2 associated X protein (Bax) (T40051), anti-B-cell CLL/lymphoma 2 (Bcl-2) (T40056), anti-autophagy related 7 (ATG7) (T57051M), anti-autophagy related 5 (ATG5) (T55766M), anti-Beclin-1 (T55092), anti-Caspase-3 (T40044), and anti-Caspase-9 (T40046) were obtained from Abmart (Shanghai, China). Anti-microtubule associated protein 1 light chain 3 beta (LC3B) (#83506), anti-alpha smooth muscle actin (α-SMA) (#19245), anti-transforming growth factor beta 1 (TGF-β1) (#3711), anti-collagen type IV alpha 1 chain (COL4A1) (#50273), and anti-β-actin (#4970) antibodies were obtained from Cell Signaling Technology (Danvers, MA, USA). Anti-sirtuin 3 (Sirt3) (sc-365175), anti-heme oxygenase 1 (HO-1) (sc-136960), and anti-superoxide dismutase 2 (SOD2) (sc-137254) antibodies were obtained from Santa Cruz Biotechnology (Dallas, TX, USA). All secondary antibodies are from Jackson ImmunoResearch (Philadelphia, PA, USA). Other chemicals and reagents were of analytical grade and were purchased locally.

### 4.2. Animals and Treatments

A total of 32 6-week-old male C57BL/6 mice were collected from the experimental animal center of Jiangsu University (Zhenjiang, China). The mice were housed in a well-controlled temperature environment (23 ± 2 °C) and subjected to a 12-h light-dark cycle, with water and food provided ad libitum. After one week of adaption to these conditions, the 32 mice were randomly divided into four groups (*n* = 8 per group): (1) The control group (given purified water as drinking water); (2) the Cd group (given purified water containing 50 mg/L Cd); (3) the MP group (given purified water containing 10 mg/L MPs of 5 μm particle size); and (4) the MP and Cd co-treatment group (given purified water containing 50 mg/L Cd and 10 mg/L of 5 μm particle size MPs). Dosage was determined according to previous studies [10,51,52]. All groups drank water ad libitum and after 3 months of contamination, all mice were weighed and anesthetized with 2% sodium pentobarbital and then sacrificed by cervical decapitation. Blood samples were taken from the ventral side of the aorta, and serum was obtained by centrifuging the samples at 2000× *g* for 15 min. The kidneys were immediately removed, and the kidney cortex was isolated. The tissues were fixed in 2.5% glutaraldehyde or 4% paraformaldehyde (PFA) or stored in a −80 °C freezer until further analysis.

### 4.3. Hematoxylin and Eosin (H&E) Staining and Histological Analysis

The kidney tissues collected from mice were fixed in 4% PFA at 4 °C for 24 h and then cut into 3 mm-thick sections. The samples were dehydrated in graded solutions of ethanol, soaked in xylene, embedded in paraffin, and sectioned to 4 μm thickness. The obtained tissue sections were assembled on slides and stained with hematoxylin and eosin (H&E) for histological analysis. All samples were observed and photographed under a Leica light microscope equipped with a digital camera (DMI3000B, Leica, Wetzlar, Germany).

### 4.4. Transmission Electron Microscopy

For transmission electron microscopy observation, small (1 mm^3^) pieces of kidney tissue from each group were fixed overnight at 4 °C in pre-chilled 2.5% glutaraldehyde. The samples were washed three times in phosphate-buffered saline (PBS) and then fixed using 1% osmium tetroxide. Thereafter, the samples were washed three times in PBS, dehydrated in an ethanol graded solution, and embedded in epoxy resin. Ultrathin sections were obtained using an ultramicrotome (EM UC7, Leica), which were stained with uranyl acetate and lead citrate. The sections were then examined using a transmission electron microscope (Tecnai 12, Philips, Holland).

### 4.5. Localization of Fluorescent Polystyrene Microspheres in Renal Tissue

To study the accumulation and localization of MPs in the kidney, we fed mice with 1 mg/mL fluorescent polystyrene microspheres for one month (0.2 mL/day via gavage). The mice were sacrificed, fresh kidney tissues were excised, and optimal cutting temperature (OCT)-embedded/fixed tissue was obtained and OCT-embedded after dehydration in a 15–30% gradient sucrose solution. The sections were stored at low temperature after completion. Sections were observed under a fluorescence microscope (TCS SP8 STED, Leica) to assess the aggregation and localization of fluorescent MPs.

### 4.6. Oxidative Stress Assessment

Fresh mouse kidney tissue samples were collected and stored at −80 °C. Catalase (CAT) activity and glutathione (GSH), malondialdehyde (MDA), and superoxide dismutase (SOD) levels were measured using commercial kits according to the manufacturer’s instructions (Nanjing Jiancheng Bioengineering Institute).

### 4.7. Western Blotting

Kidney tissue was ground and centrifuged to extract the precipitate and the tissue was lysed on ice for 15 min using Radioimmunoprecipitation assay (RIPA) lysis solution containing protease inhibitors. After ultrasonic lysis on ice for 15 min, the supernatant was collected by centrifugation at 12,000× *g* for 10 min at 4 °C and the protein concentration was quantified using the bicinchoninic acid (BCA) method. Equal amounts of proteins were separated using sodium dodecyl sulfate polyacrylamide gel electrophoresis (SDS-PAGE) and then the proteins were electrotransferred to polyvinylidene difluoride (PVDF) membranes. After blocking in 5% skim milk powder for 2 h at room temperature, the membranes were incubated with primary antibodies overnight at 4 °C. The next day, the membranes were washed three times with Tris-buffered saline-Tween 20 (TBST) on a shaker for 10 min each time and then incubated for 2 h at room temperature with the appropriate horseradish peroxidase-conjugated secondary antibody. After incubation, the membranes were washed three times using TBST for 10 min each time on a shaker and then detected using enhanced chemiluminescence. Immunoreactive protein levels were analyzed using Image Lab software (National Institutes of Health, Bethesda, MD, USA). The density of each band was normalized to its corresponding loading control (β-actin).

### 4.8. Immunohistochemical Analysis

Kidney tissue was fixed in 4% PFA and embedded in paraffin. The embedded tissue was cut into 2 μm sections. For immunohistochemical (IHC) staining, the sections were incubated with anti-Bax antibodies (mouse, 1:250, Abcam, Cambridge, MA, USA); anti-cytochrome C (Cytc) antibodies (mouse, 1:250, Abcam), and anti-Caspase-3 antibodies (mouse, 1:100, Abcam). Slides were scanned and imaged using an SCN400 slide scanner (Leica). The quantitative analysis of the images was measured by Image-Pro Plus 6.0 software.

### 4.9. Sirius Red Staining

Kidney tissue was fixed in 4% PFA and embedded in paraffin. The embedded tissue was cut into 2 μm sections and stained with Sirius Red. Sirius Red staining kits were purchased from Abcam. After the sections were dewaxed, stained, dehydrated, and made transparent, they were sealed for observation. Sirius Red staining was used to determine collagen deposition as a measure of the extent of liver fibrosis. The slides were scanned and imaged using a SCN400 slide scanner (Leica). The quantitative analysis of the images was measured by Image-Pro Plus 6.0 software.

### 4.10. Masson Staining

A 4% PFA-fixed kidney tissue block was cut to 0.5 cm^3^ size for paraffin embedding, followed by dewaxing and rinsing in distilled water. Tissues were stained with hematoxylin for 5–10 min and then washed with distilled water. The tissue was then stained with Lichon Red acidic magenta staining solution for 5–10 min, followed by a period of immersion in 2% glacial acetic acid aqueous solution and 1% phosphomolybdic acid aqueous solution for 3–5 min. After drying, the stained tissue was incubated with Aniline Blue solution for 5 min. After further incubation in 0.2% glacial acetic acid for 1 min, the pieces were quickly washed and dehydrated with 95% ethanol and anhydrous ethanol. Finally, the pieces were sectioned, infiltrated with xylene, dried, and sealed with neutral glue. The sections were then viewed under a microscope (Leica). The quantitative analysis of the images was measured by Image-Pro Plus 6.0 software.

### 4.11. Statistical Analyses

The results are expressed as the mean ± standard deviation (SD) of at least three independent experiments. Significance was calculated by one-way analysis of variance (ANOVA) using SPSS 26.0 software (IBM Corp., Armonk, NY, USA), followed by Tukey’s test. The results were considered statistically significant at a threshold of *p* < 0.05.

## 5. Conclusions

We revealed the localization and toxicological effects of MPs in a Cd-mediated mouse kidney injury model. To investigate the damaging effects of MPs and cadmium on the kidney, we applied 10 mg/L particle 5 μm MPs and 50 mg/L CdCl_2_ singly and in combination in an in vivo assay. The results showed that polystyrene particles of 5 μm diameter could enter the systemic circulation and accumulate in the kidneys of mice where they induced severe biological responses. The kidneys of mice exposed to MPs and Cd exhibited a state of oxidative stress, autophagy, apoptosis, and fibrosis, leading to kidney damage, alteration of kidney tissue structure, and ultimately, nephrotoxicity. Hence, these findings provide further evidence for the threat of MPs and their adsorbed heavy metals.

## Figures and Tables

**Figure 1 ijms-23-14411-f001:**
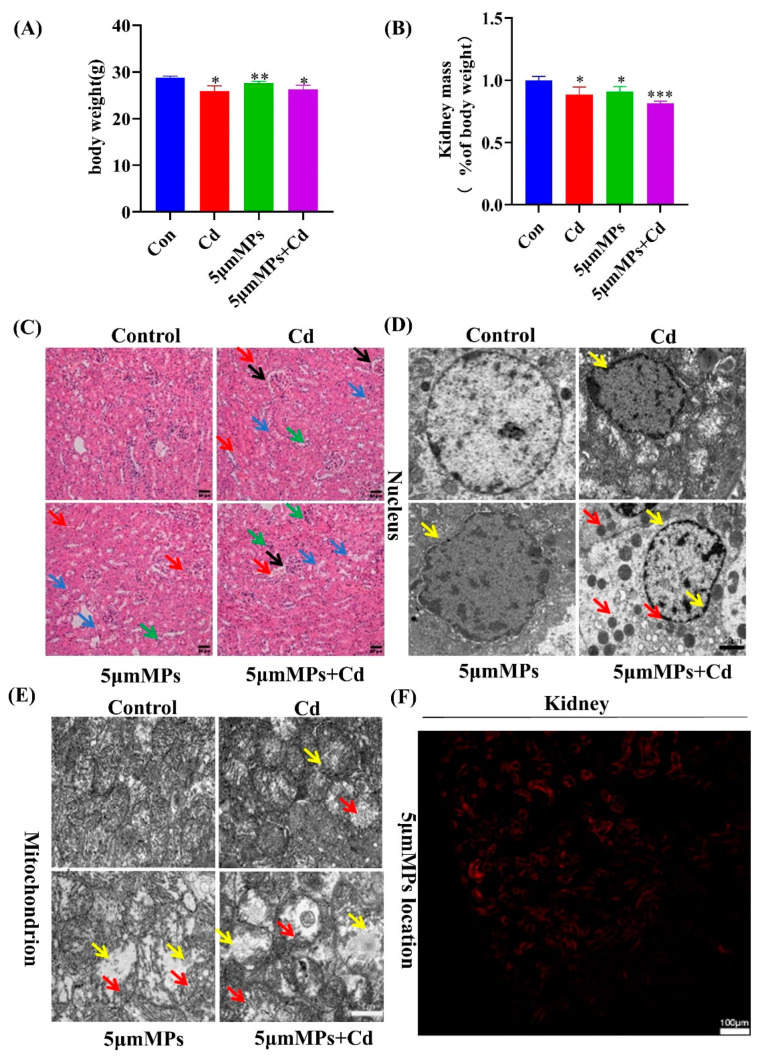
Microplastics (MPs) aggravate cadmium (Cd)-induced kidney injury. Mice were treated with Cd (50 mg/L) and/or 5 μm MPs (10 mg/L) for 90 days. (**A**) Body weight (g), weight of the animal before dissection. (**B**) Kidney weight (% body weight) (**C**) Hematoxylin and eosin (H&E) staining of kidney samples from the mice in each group. a: Glomerulus, b: Renal tubule. A period of 90 days of Cd exposure induced dilatation of renal capsule (black arrow), necrosis and shedding of the renal tubular epithelium (blue arrow), and increased red blood cell (red arrow) and monocyte infiltration (green arrow). Treatment with MPs aggravated Cd-induced kidney injury in mice. Scale bars: 50 µm. Representative electron microscope images showing the nuclei (**D**) and mitochondria (**E**) in the kidneys of mice in each group. Damage to nuclear, as evidenced by nuclear depression and deformation (yellow arrow) and lipid droplets (red arrow). Damage to mitochondria, as evidenced by significant vacuolization (yellow arrow) and cristae disruption (red arrow). Scale bars: 20 µm and 10 µm. (**F**) Bioaccumulation of MPs in mouse kidneys was determined using laser scanning confocal microscopy (5 μm). Red fluorescence indicates the location of MPs. Scale bars: 100 µm. Data are presented as means ± SD from three independent experiments. * *p* < 0.05, ** *p* < 0.01, *** *p* < 0.001 compared to the control group.

**Figure 2 ijms-23-14411-f002:**
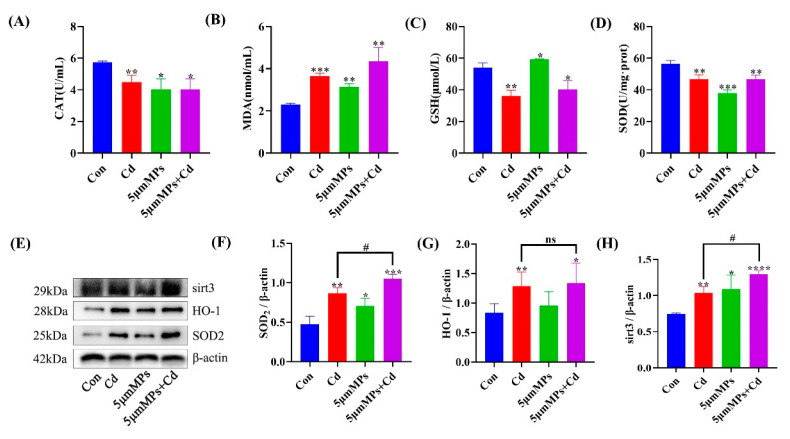
Effect of MPs and Cd on oxidative stress in mouse kidneys. Mice were treated with Cd (50 mg/L) and/or 5 μm MPs (10 mg/L) for 90 days. (**A**–**D**): The levels of catalase (CAT) activity, glutathione (GSH), and malondialdehyde (MDA were evaluated in the mouse serum. (**E**–**H**): the levels of sirtuin 3 (Sirt3), heme oxygenase 1 (HO-1), and superoxide dismutase 2 (SOD2) were detected using western blotting. (**E**): Representative western blot image; (**F**–**H**): Quantitative analysis. Data are presented as means ± SD from three independent experiments. * *p* < 0.05, ** *p* < 0.01, *** *p* < 0.001, **** *p* < 0.0001 compared to the control group; # *p* < 0.05 compared to the Cd group.

**Figure 3 ijms-23-14411-f003:**
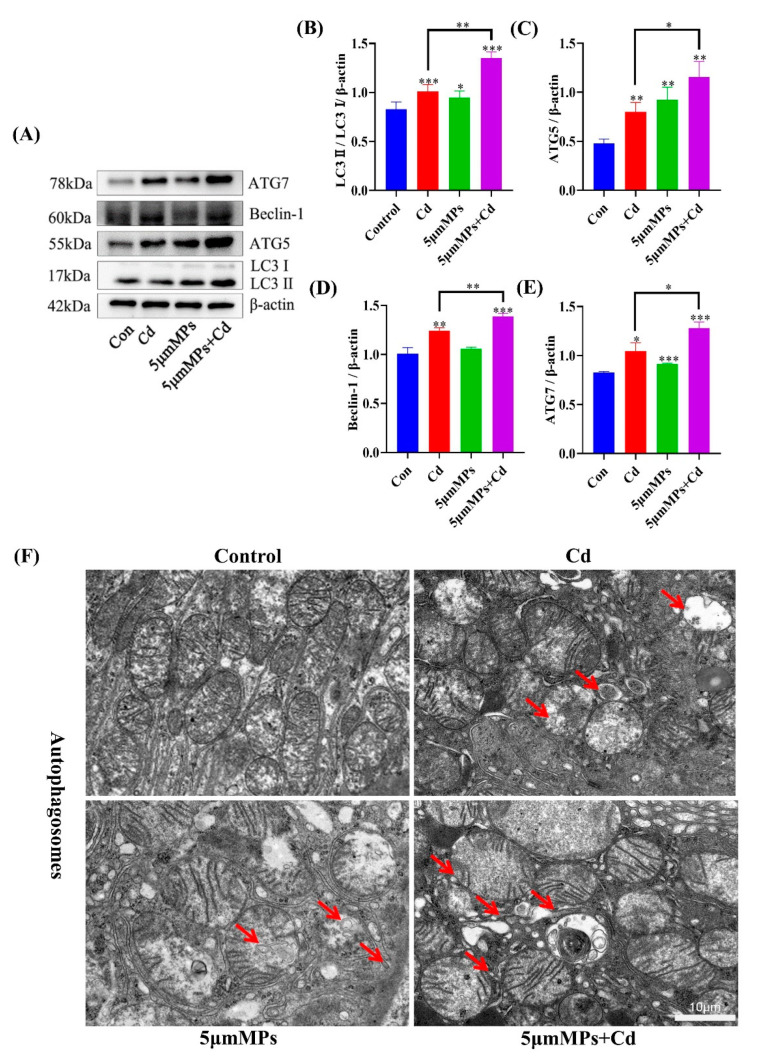
Effect of MPs and Cd on autophagy in mouse kidneys. Mice were treated with Cd (50 mg/L) and/or 5 μm MPs (10 mg/L) for 90 days. (**A**–**E**): The levels of LC3, ATG5, Beclin-1, andATG7 were detected by Western blot. (**A**): representative Western blot image; (**B**–**E**): quantitative analysis. (**F**): Representative electron microscope images show the autophagosomes (red arrow) in the kidneys of mice in each group. Scale bars: 1 µm. Data are presented as means ± SD from three independent experiments. * *p* < 0.05, ** *p* < 0.01, *** *p* < 0.001 compared to the control group.

**Figure 4 ijms-23-14411-f004:**
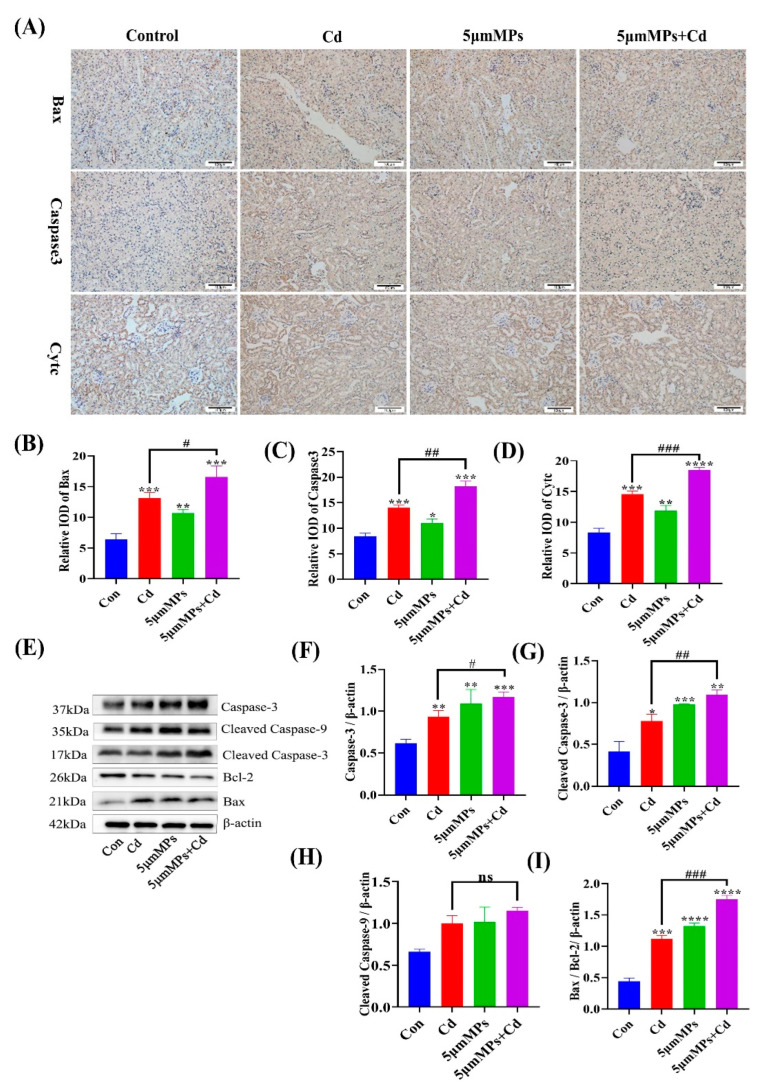
Effect of MPs and Cd on apoptosis in mouse kidneys. Mice were treated with Cd (50 mg/L) and/or 5 μm MPs (10 mg/L) for 90 days. (**A**–**D**): BCL2 associated X protein (Bax), Caspase-3, and cytochrome C (Cytc) levels in different groups assessed using an immunohistochemical (IHC) assay. Scale bars: 50 µm. (**C**,**D**): Quantitative analysis. (**E**–**I**). The levels of Bax, B-cell CLL/lymphoma 2 (Bcl-2), Caspase-3, Cleaved Caspase-3, and Cleaved Caspase-9 were detected by western blotting. (**E**): Representative western blot image; (**F**–**I**): Quantitative analysis. Data are presented as means ± SD from three independent experiments. * *p* < 0.05, ** *p* < 0.01, *** *p* < 0.001, **** *p* < 0.0001 compared to the control group; # *p* < 0.05, ## *p* < 0.01, ### *p* < 0.001 compared to the Cd group.

**Figure 5 ijms-23-14411-f005:**
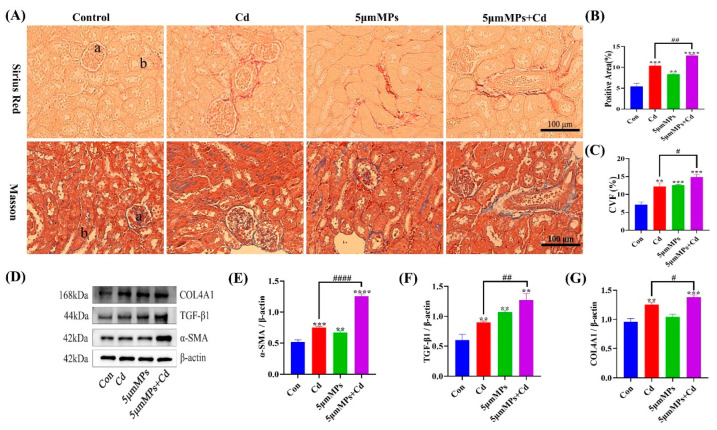
Effect of MPs and Cd on fibrosis in mouse kidneys. (**A**–**C**): Representative images of Sirius Red staining and Masson staining of the kidney after MP and/or Cd exposure. a: Glomerulus, b: Renal tubule. (**B**,**C**): Quantitative analysis. (**D**–**G**): The levels of collagen type IV alpha 1 chain (COL4A1), transforming growth factor beta 1 (TGF-β1), and alpha smooth muscle actin (α-SMA) were detected using western blotting. (**D**): Representative western blot image; (**E**–**G**): Quantitative analysis. Data are presented as means ± SD from three independent experiments. ** *p* < 0.01, *** *p* < 0.001, **** *p* < 0.0001 compared to the control group; # *p* < 0.05, ## *p* < 0.01, #### *p* < 0.0001 compared to the Cd group.

## Data Availability

The data presented in this study are available on request from the corresponding author Z.L.

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
