# Peer review of "Microplastics Exacerbate Cadmium-Induced Kidney Injury by Enhancing Oxidative Stress, Autophagy, Apoptosis, and Fibrosis"

_ijms, 2022, doi:10.3390/ijms232214411_

Round 1

Reviewer 1 Report

The manuscript by Hui Zou et al. deals of in vivo combined effect of microplastic particles and cadmium on mouse kidney. This is an interesting and very actual topic, given the increased environmental pollution and consequential risk for ecosystem and human health. However, this work is not totally convincing, as results are not clearly presented and described, thus their biological significance remains uncertain.

Specific comments:

Abstract:

Authors should specify some details about the experimental design, for example the size of microplastics employed, time and concentration of treatments, etc. I suggest to enpathyze conclusions regarding possible implication for mammals health.

Results:

The size of particles employed has to be clearly indicated in the text of results. The rationale for the use of reported concentrations of both MPs and Cd has to be indicated.

Figure 1 and its legend has to been improved

-Specify the time of body weight.

-Arrows in the panel C are almost invisible and colours are different from those cited in the text.

-Also scale bars have to be clearly reported for each panel and in the legend (in next figures too).

-Clarify the meaning of “b a” in panel C.

-In panel D change nucleolus with nucleus.

-Add arrows to highlight changes in the TEM images too.

-In panel F the description of each image is missing.

-In the legend it is indicated that only three experiments have been performed to measure mean and DS. Why Authors did not report values for all sacrificed animals?

Figure 3

Specify which form of LC3 has been detected. Use arrows to indicate increasing autophagosomes. In the legend change 5 m M with 5 mm.

Figure 4

It’s not possible to establish significative differences in antigen expression in HIC panels. Differences are only clearly evident with the respect to controls panels. Moreover, control panel of caspase 3 has a different magnification. Western blots are not convincing: lines of C3 are too much similar to lines of CC3, in form and intensities. Original blots are not convincing too and there are not any replicate for comparison. Thus, the biological significance of these experiments is not clear.

Figure 5

Same comment of figure 4 for HIC panels.

Discussion should be better highlight previous data on kidney toxicity in the presence microparticle of 5 mM diameters and previous data supporting that there is a physical interaction between Cd and microparticles.

Author Response

We have revised the manuscript after considering the reviewers’ comments carefully and have made revision which marked in red in the paper. We have tried our best to revise our manuscript according to the comments. Attached is the point-by–point response to all the comments and suggestions of the reviewers and your editorial comments. We would like to express our great appreciation to you and reviewers for comments on our paper. If you have any question about the manuscripts, please let me know.

Reviewer 2 Report

Chronic Cadmium exposure leads to the accumulation of metal in the human body and causes irreversible organ damage the most affected are the kidney. Any intoxication characterized by the generation of free radicals and high levels of oxidative stress, like Cd exposure is no exception.

In the current study, the authors evaluate the nephrotoxic effects of Microplastic particles (MPs) on Cadmium-induced nephrotoxicity in a mouse model. They investigate the possibility of enhancing the toxic effect of Cd and MPs (less than 5 μm in diameter) in kidneys. At the same time, the article emphasizes the important role of exogenous MPs/Cd in the generation of free radicals (ROS), inhibition of the antioxidant system, inducing apoptosis, autophagy, and subsequent fibrosis in mouse kidney cells, as basic elements in renal pathology and kidney dysfunction. 

The authors used a large number of specific methods that fully correlate with the purpose of the present study. The results show that the microspheres accumulated in the renal tubular epithelium, which assumes that 5 μm MPs can exacerbate Cd-induced kidney injury and co-treatment MPs/Cd increased the levels of general and renal oxidative stress in mice.

The study was systematic and organized. Data analyses and results appeared to be logical, interesting, well-presented, and consistent with the discussion and conclusions. 

Author Response

(The authors gave the same response as above.)

Reviewer 3 Report

Dear Editors:

The authors evaluated the mechanism of oxidative stress, autophagy, apoptosis, and fibrosis in the microplastics and cadmium-induced kidney injury. The experiment design was good, results and conclusion were clear and reliable. But the manuscript preparation and writing were poor, especially the figures. Therefore, I suggested accepting the manuscript after the Minor Revision.

1.    The resolution of the figures in the manuscripts needs to improve. E.g. figure 5A, I didn’t see clearly of the glomerulus structure.

The figure 3 E, the resolution of electron microscope images in 5 μM MPs group was different with the others.

Figure 4A, the magnification in the Caspase-3 group is different with the others.  Be very careful about this, otherwise, the figures will make people wonder if you cook the book.

2.    All the figures legends need to be revised, the authors should add detailed information e.g. the animal’s numbers, the drug administration routes, the magnification fold of the staining, and so on.

3.    Ask a native American to help you with the writing, e.g. “2.3. Effect of co-treatment with MPs and Cd on autophagy in mouse kidney tissue”. Co-treatment is appropriate when coordination between the two disciplines will benefit the patient.  We say in mouse kidney instead of in mouse kidney tissue. Singular and plural, grammar.

4.    Ask some pathologist to help you analyze the staining, I think most of the histological analysis was not accurate based on my knowledge.

5.    On page 2 of 16, in the introduction parts, you can’t write the results.

Author Response

(The authors gave the same response as above.)

Round 2

Reviewer 1 Report

See attached file.

Author Response

We have revised the manuscript after careful consideration of the reviewers' comments and have marked the revisions in red throughout the paper. Every effort has been made to revise our manuscript in response to comments. Include point-by-point responses to all comments and suggestions from the reviewers, as well as your editorial comments. We greatly appreciate your comments and the reviewers' comments on our paper. Please let me know if you have any questions about the manuscript.
